# Coronary Artery Bypass Graft Surgery Brings Better Benefits to Heart Failure Hospitalization for Patients with Severe Coronary Artery Disease and Reduced Ejection Fraction

**DOI:** 10.3390/diagnostics12092233

**Published:** 2022-09-16

**Authors:** Yu-Wen Chen, Wei-Chieh Lee, Hsiu-Yu Fang, Cheuk-Kwan Sun, Jiunn-Jye Sheu

**Affiliations:** 1Division of Thoracic and Cardiovascular Surgery, Department of Surgery, Kaohsiung Chang Gung Memorial Hospital, Chang Gung University College of Medicine, Kaohsiung 83301, Taiwan; 2Institute of Clinical Medicine, College of Medicine, National Cheng Kung University, Tainan 70101, Taiwan; 3Division of Cardiology, Department of Internal Medicine, Chi Mei Medical Center, Tainan 71004, Taiwan; 4Division of Cardiology, Department of Internal Medicine, Kaohsiung Chang Gung Memorial Hospital, Chang Gung University College of Medicine, Kaohsiung 83301, Taiwan; 5Department of Emergency Medicine, E-Da Hospital, Kaohsiung 83301, Taiwan; 6School of Medicine for International Students, College of Medicine, I-Shou University, Kaohsiung 83301, Taiwan

**Keywords:** severe coronary artery disease, reduced ejection fraction, coronary artery bypass graft surgery, percutaneous coronary intervention, heart failure hospitalization

## Abstract

Objectives: We compared the outcomes between percutaneous coronary intervention (PCI) and coronary artery bypass graft surgery (CABG) for revascularization in patients with reduced ejection fraction (EF) and severe coronary artery disease (CAD). Methods: Between February 2006 and February 2020, a total of 797 patients received coronary angiograms due to left ventricular EF ≤ 40% at our hospital. After excluding diagnoses of dilated cardiomyopathy, valvular heart disease, prior CABG, acute ST-segment myocardial infarction, and CAD with low Synergy between PCI with Taxus and Cardiac Surgery (SYNTAX) score (≤22), 181 patients with severe coronary artery disease (CAD) with SYNTAX score >22 underwent CABG or PCI for revascularization. Vascular characteristics as well as echocardiographic data were compared between CABG (*n* = 58) and PCI (*n* = 123) groups. Results: A younger age (62 ± 9.0 vs. 66 ± 12.1; *p* = 0.016), higher new EuroSCORE II (8.6 ± 7.3 vs. 3.2 ± 2.0; *p* < 0.001), and higher SYNTAX score (40.5 ± 9.8 vs. 35.4 ± 8.3; *p* < 0.001) were noted in the CABG group compared to those in the PCI group. The CABG group had a significantly higher cardiovascular mortality rate at 1-year (19.6% vs. 5.0%, *p* = 0.005) and 3-year (25.0% vs. 11.4%, *p* = 0.027) follow-ups but a lower incidence of heart failure (HF) hospitalization at 1-year (11.1% vs. 28.2%, *p* = 0.023) and 3-year (3.6% vs. 42.5%, *p* = 0.001) follow-ups compared to those of the PCI group. Conclusions: Compared with PCI, revascularization with CABG was related to a lower incidence of HF hospitalization but a worse survival outcome in patients with severe CAD and reduced EF. CABG-associated reduction in HF hospitalization was more notable when SYNTAX score ≥33.

## 1. Background

Ischemic heart disease has been the leading cause of death worldwide in the past decades [1]. Previous studies have shown a better survival outcome in patients with ischemic heart disease undergoing revascularization than those receiving medical treatments [2,3]. For patients with severe coronary artery disease (CAD) and reduced ejection fraction (EF), coronary artery bypass graft surgery (CABG) or percutaneous coronary intervention (PCI) are options for revascularization [4].

Morbidity and mortality remain high in the population of heart failure (HF) with reduced EF despite recent developments in medical care [5,6]. Although heart failure with reduced EF accounts for approximately 50% of cases [7,8], there is no consensus about the preference to receive PCI or CABG for patients with severe CAD and reduced EF in current medical guidelines. In some studies, patients who received CABG had better overall survival compared to those who underwent PCI [9,10], though the others conclude that there is no significant difference in both groups [11,12]. In such a population, the evaluation of the ischemic area and involved vessels and the healthy collateral vessel are very important and the methods of PCI or CABG also have an impact on the outcomes [13,14,15,16]. The decompensated status of HF with reduced EF and post-PCI or CABG stunning cardiomyopathy also plays an important role in this issue. 

There are only a few large, randomized control studies that compared the two revascularization strategies in this clinical setting [9,17]. Accordingly, we conducted this study to investigate the outcomes and improvements in ventricular performance receiving either PCI or CABG in patients with severe CAD and reduced EF. 

## 2. Methods

### 2.1. Ethical Statement

The protocol and procedures of this study were in accordance with the Declaration of Helsinki and were approved by the Institutional Review Board (IRB) of Kaohsiung Chang Gung Memorial Hospital. Informed consent was waived due to the retrospective nature of the study (IRB No.: 202100929B0; approval date: 25 June 2021).

### 2.2. Study Protocol and Population

Between February 2006 and February 2020, a total of 797 patients with HF with reduced EF receiving coronary angiography were recruited. The medical records of all patients (age > 18) with severe heart failure (defined as left ventricular EF ≤ 40%) regardless of gender undergoing either PCI by a single intervention cardiologist or CABG by a single cardiovascular surgeon with pre-procedural coronary angiograms at a tertiary referral center were retrospectively reviewed for demographic (i.e., age, gender, body mass index) and clinical (e.g., comorbidities, information, medications, cardiac functional status) information as well as coronary complexity based on the SYNTAX (Synergy between PCI with Taxus and Cardiac Surgery) score, and echocardiographic findings were recorded. Patients with (1) dilated cardiomyopathy or severe valvular heart disease with and without surgery, (2) prior history of CABG, (3) acute ST-segment myocardial infarction, (4) CAD with a SYNTAX score ≤ 22, and (5) emergent CABG for coronary perforation were excluded from the present study. The enrolled patients were severe CAD (SYNTAX score > 22), who need PCI or CABG for ischemic cardiomyopathy and HF with reduced EF. The clinical condition, vascular characteristics, and echocardiographic data were compared between groups. PCI were classified into two subgroups: complete revascularization (PCI-CR) and incomplete revascularization (PCI-ICR) to evaluate the effect of CR.

### 2.3. Echocardiography

All patients underwent echocardiographic examination by using a Philips IE33 before the procedure and at least annually thereafter in the absence of clinical events. Additional examinations were performed for patients at the onset of heart failure or other cardiovascular events. Left atrial (LA) dimension was measured by the M-mode. Left ventricular ejection fraction (LVEF), left ventricular end-diastolic volume (LVEDV) and left ventricular end-systolic volume (LVESV) were quantified by the M-mode and corrected by the two-dimensional guided biplane Simpson’s method of disc measurements. Follow-up echocardiography was performed at least annually thereafter in the absence of clinical events or at the onset of heart failure or cardiovascular events or more frequently among more complex conditions.

### 2.4. Definitions

HF hospitalization was defined as the occurrence of HF events according to a New York Heart Association functional class ≥III in the absence of other alternative diagnoses. Cardiovascular (CV) mortality was defined as sudden death related to arrhythmias, HF, and myocardial infarction. All-cause mortality was defined as death related to any cause, such as sudden death with undefined reasons, natural course, sepsis, malignancy, HF, and cardiovascular disease. For patients undergoing PCI, complete revascularization (PCI-CR) was defined as successful revascularization of all stenotic coronary arteries, while incomplete revascularization (PCI-ICR) referred to unsuccessful revascularization of any vessel after the procedure.

### 2.5. Study Endpoints

The study endpoints were overall mortality (i.e., in-hospital, cardiovascular and all-cause), recurrent myocardial infarction, sudden death or ventricular arrhythmia, HF hospitalization and stroke. 

### 2.6. Statistical Analysis

Continuous variables are presented as mean ± standard deviation (SD) and categorical data are expressed as absolute values with percentages for normally distributed parameters and showed median and interquartile range for non-normally distributed parameters. Baseline patient characteristics of the two groups were compared with a 2-sample t-test for continuous variables and with a Chi-square test or Fisher’s exact test for categorical variables. Cox regression analyses on heart failure hospitalization were performed to determine the HR between groups. The patient in the CABG group was set as an HR of 1. Mortality rates and HF readmission rates were calculated using the Kaplan–Meier method with the log-rank test. A probability value less than 0.05 was considered statistically significant in this study. Statistical analysis was performed using statistical software (SPSS for Mac, Version 26, IBM. Corp., Armonk, NY, USA).

## 3. Results

### 3.1. Baseline Characteristics

Of the 797 adult patients receiving coronary angiograms due to left ventricular EF ≤40% at our hospital during the study period, 616 were excluded because of dilated cardiomyopathy or severe valvular heart disease with and without surgery (*n* = 256), prior history of CABG (*n* = 69), acute ST-segment myocardial infarction (*n* = 103), CAD with low SYNTAX score ≤22 (*n* = 185) and emergent CABG for coronary perforation (*n* = 3). Finally, 181 patients with severe CAD and SYNTAX score >22 were recruited for the current study (Figure 1). Of the 181 patients, 58 underwent CABG and 123 received PCI for revascularization.

Baseline characteristics of study patients are listed in Table 1. Patients in the CABG group were younger than those in the PCI group (CABG vs. PCI; 62 ± 9.0 vs. 66 ± 12.1; *p* = 0.016) and PCI-CR group (CABG vs. PCI-CR; 62 ± 9.0 vs. 66 ± 12.1; *p* = 0.049), but there were no significant differences in gender and body mass index among the three groups. The incidences of comorbidities including hypertension, diabetes mellitus, advanced chronic kidney disease and smoking did not differ among the three groups (CABG vs. PCI and CABG vs. PCI-CR). On the other hand, a higher prevalence of prior PCI history (CABG vs. PCI; 46.6% vs. 19.5%; *p* < 0.001; CABG vs. PCI-CR; 46.6% vs. 17.6%; *p* = 0.001) and a poorer New York Heart Association functional class (≥3) (CABG vs. PCI; 81.0% vs. 56.1%; *p* = 0.001; CABG vs. PCI-CR; 46.6% vs. 17.6%; *p* = 0.005) were noted in the CABG group compared to those in the PCI subgroups. The CABG group also had higher New EuroSCORE II score (CABG vs. PCI; 6.25 (4.15−8.29) vs. 2.73 (2.48−3.26); *p* < 0.001; CABG vs. PCI-CR; 6.25 (4.15−8.29) vs. 2.81 (2.40−3.64); *p* < 0.001), and SYNTAX score (CABG vs. PCI; 40.5 ± 9.8 vs. 35.4 ± 8.3; *p* < 0.001; CABG vs. PCI-CR; 40.5 ± 9.8 vs. 35.3 ± 7.9; *p* = 0.001), and prevalence of high SYNTAX score ≥33 (CABG vs. PCI; 79.3% vs. 57.7%; *p* = 0.005; CABG vs. PCI-CR; 79.3% vs. 60.8%; *p* = 0.025) than those in the PCI group. However, the prevalence of left main disease, two-vessel disease, and three-vessel disease did not differ among the three groups (CABG vs. PCI and CABG vs. PCI-CR). With respect to medications, the prevalence of patients in the CABG group using angiotensin-converting enzyme inhibitor (ACEI)/angiotensin II receptor blocker (ARB)/angiotensin receptor–neprilysin inhibitor (ARNI), β-blocker, and diuretics was lower than that in the PCI group despite a higher prevalence of spironolactone use in the former.

Echocardiographic examination before the procedures showed no significant differences in functional parameters between the CABG group and PCI group, except for a higher prevalence of patients with aortic regurgitation >2 in the PCI group (CABG vs. PCI; 11.4% vs. 1.7 %, *p* = 0.039) (Table 1). After revascularization, there were also no significant differences in chamber size and valvular disease between the CABG group and PCI group despite a non-significantly higher LVEF in the CABG group (CABG vs. PCI; 45.4 ± 13.8% vs. 42.8 ± 15.3%, *p* = 0.288).

The percentage of patients with LVESV improvement >10% was significantly higher in the CABG group than that in the PCI group (67.3% vs. 50.0%, respectively, *p* = 0.041). Nevertheless, the difference in the percentage of patients with LVESV improvement >10% was non-significant between the CABG and the PCI-CR groups (67.3% vs. 52.5%, respectively, *p* = 0.127). Although there were significant elevations in LVEF on three-year follow-up compared to their respective baseline values in both groups (CABG: 45.4 ± 13.8% vs. 31.2 ± 6.8%, *p* < 0.001; PCI: 42.8 ± 15.3% vs. 31.6.4 ± 7.1%, *p* < 0.001; respectively), there was no significant difference in the degree of improvement between the two groups at the end of three-year follow-up (CABG vs. PCI; 13.5% (7.9–20.1%) vs. 10.5% (5.0–17.4%); *p* = 0.233). The percentage of patients with improvements in mean LVEF >10% also did not differ between the two groups (CABG vs. PCI; 61.5% vs. 52.9%; *p* = 0.391). 

### 3.2. Clinical Outcomes of the Study Patients

During hospitalization, the mortality, the incidence of acute kidney injury, and the need for post-procedural hemodialysis did not differ between patients in the CABG and PCI groups (Table 2).

During the one-year follow-up period, compared to those in the PCI and the PCI-CR groups, the CABG group had significantly higher incidences of sudden death/ventricular arrhythmia (CABG vs. PCI; 13.5% vs. 2.6%; *p* = 0.011; CABG vs. PCI-CR; 13.5% vs. 1.5%, *p* = 0.021), CV mortality (CABG vs. PCI; 19.6% vs. 5.0%; *p* = 0.005; CABG vs. PCI-CR; 19.6% vs. 5.6%, *p* = 0.024) and all-cause mortality (CABG vs. PCI; 22.4% vs. 8.1%; *p* = 0.005; CABG vs. PCI-CR; 22.4% vs. 8.1%, *p* = 0.025). On the other hand, a significantly lower incidence of HF hospitalization was noted in the CABG group compared to that in the PCI and the PCI-CR groups (CABG vs. PCI; 11.1% vs. 28.2%; *p* = 0.023; CABG vs. PCI-CR; 11.1% vs. 28.2%, *p* = 0.037). During the three-year follow-up period, a lower incidence of the need for revascularization was noted in the CABG group when compared with that in the PCI group (6.9% vs. 22.0%, respectively; *p* = 0.011). However, a significantly higher incidence of sudden death/ventricular arrhythmia was found in the CABG group compared to that in the PCI group (19.2% vs. 8.3%, respectively; *p* = 0.044).

Consistent with the findings during one-year follow-up, the CABG group had a significantly lower incidence of HF hospitalization (CABG vs. PCI; 13.6% vs. 42.5%; *p* = 0.001; CABG vs. PCI-CR; 13.6% vs. 39.7%, *p* = 0.003). Hazard ratio of 3-year HF hospitalization compared to CABG were 3.868 (*p* = 0.002) in PCI group and 3.553 (*p* = 0.005) in PCI-CR group (Table 3). The difference was only found on patients with Syntax score ≥33. Similarly, a significantly higher incidence of CV mortality was noted in the CABG group compared to that in the PCI group (CABG vs. PCI; 25.0% vs. 11.4%; *p* = 0.027) but there was no difference in this parameter between the CABG and PCI-CR groups (25.0% vs. 14.1%, respectively, *p* = 0.170). *: statistically significant.

### 3.3. Kaplan–Meier Analysis of All-Cause Mortality and Heart Failure Hospitalization at One-Year and Three-Year Follow-Ups

Although the CABG group had a higher incidence of all-cause mortality than that in the PCI group at one-year follow-up (log-rank *p* = 0.008), there was no significant difference between the two groups at three-year follow-up (log-rank *p* = 0.088) (Figure 2A). Similar results were noted when the all-cause mortality associated with CABG was compared with that of the PCI-CR and PCI-ICR groups at one-year (log-rank *p* = 0.030) and three-year (log-rank *p* = 0.232) follow-ups (Figure 2B).

The incidences of HF hospitalization were significantly lower in the CABG group than those in the PCI groups at both one-year (log-rank *p* = 0.019) and three-year (log-rank *p* < 0.001) follow-ups (Figure 2C). When the PCI group was separated into PCI-CR and PCI-ICR groups, there was no significant difference among the three groups at one-year follow-up (log-rank *p* = 0.062) but the CABG group still had a significantly lower incidence of HF hospitalization at three-year follow-up than that in the two PCI subgroups (log-rank *p* = 0.002) (Figure 2D).

### 3.4. Kaplan–Meier Analysis of All-Cause Mortality and Heart Failure Hospitalization between CABG and PCI Groups by SYNTAX Score at Follow-Ups

Subgroup analysis on patients with complex coronary artery lesions (i.e., SYNTAX score ≥ 33) demonstrated a better survival outcome for those in the PCI group at one-year follow-up (log-rank *p* = 0.012) but without difference between the CABG and the PCI groups at three-year follow-up (log-rank *p* = 0.459) (Figure 3A). Subgroup analysis focusing on patients with moderate complexity of coronary artery lesions (i.e., SYNTAX score between 33 and 22) showed no significant difference in all-cause mortality between the two groups at one-year and three-year follow-ups (log-rank *p* = 0.942; log-rank *p* = 0.061, respectively) (Figure 3B).

In respect of the impact of coronary artery lesion complexity on HF hospitalization. subgroup analysis on patients with SYNTAX score ≥33 revealed a significantly lower incidence of HF hospitalization in the CABG group than that in the PCI group at both one-year and three-year follow-ups (log-rank *p* = 0.002; log-rank *p* < 0.001, respectively) (Figure 3C). Similar findings were noted in patients with SYNTAX score between 33 and 22 at one-year and three-year follow-ups (log-rank *p* = 0.506; log-rank *p* = 0.970, respectively) (Figure 3D).

## 4. Discussion

### 4.1. The Significance of Our Study

The choice of an optimal revascularization strategy for patients with severe CAD and compromised EF remains controversial. To date, there are no guidelines recommending either PCI or CABG for this patient population [18,19]. The current study is the first investigation focusing on the comparison of short- and long-term outcomes between patients undergoing the two interventional approaches. Our results showed a significantly lower incidence of HF hospitalization in patients receiving CABG at three-year follow-up (13.6%) compared to that in the PCI group or PCI-CR group (42.5% and 39.7%, respectively), particularly in the population with a high SYNTAX score (9.1%). On the other hand, the current study revealed a higher all-cause mortality at a one-year follow-up in the CABG group (22.4%) compared to that in the PCI group (8.1%), especially in patients with a high SYNTAX score (i.e., ≥33) (27.3% vs. 7.4%).

A previous cohort study including 3584 patients with three-vessel and/or left main disease demonstrated a higher incidence of readmission for heart failure in patients with impaired LV systolic function (i.e., LVEF < 50%) undergoing PCI than that in those receiving CABG (24% vs. 15%, Hazard Ratio = 2.22, *p* < 0.01). However, there was no significant difference in the incidence of readmission between the PCI and CABG groups (8% vs. 6%, respectively; Hazard Ratio = 1.39, *p* = 0.11) for patients with preserved LV systolic function (i.e., LVEF < 50%) receiving either of the two procedures [17]. Another recent study including 4794 patients with reduced LVEF also reported consistent results (PCI vs. CABG = 25.8% vs. 20.1%, Hazard Ratio = 1.5, *p* < 0.01) [9]. Therefore, the findings of those studies were consistent with those of the present study (PCI vs. CABG; 42.5% vs. 13.6%; Hazard Ratio = 3.868, *p* = 0.002). In one metaanalysis of severely reduced EF, CABG decreases the risk of mortality, M and revascularization than PCI, but may increase the short-term and long-term risk of stroke [20].

### 4.2. Revascularization Strategy for the Patients with CAD and Reduced EF

The growth of the aging population worldwide has contributed to a global rise in cardiovascular deaths in recent years [21]. Importantly, the burden of HF has increased with approximately 50% of patients showing a reduced EF [5]. However, current guidelines only recommend that CABG may be considered in patients with ischemic heart disease and severe LV systolic dysfunction without mentioning the priority of revascularization strategy [22]. A previous study has reported a lower risk of mortality associated with CABG and PCI than that with medical treatment alone [23]. Consistently, a large database study reported higher rates of mortality and major adverse cardiovascular events in patients receiving PCI compared with those undergoing CABG, but this study did not provide the prevalence of HF hospitalization after revascularization between the two groups [9]. The SYNTAX trial revealed a significantly higher prevalence of major adverse cardiac or cerebrovascular events and repeated revascularization in patients undergoing PCI, especially in those with high SYNTAX scores [24]. However, the SYNTAX trial did not mention HF hospitalization and the change in LVEF after revascularization. Currently, the optimal revascularization strategy for severe CAD (i.e., SYNTAX score ≥ 33) remains controversial. 

In our study, although the all-cause mortality in the PCI group was significantly lower than that in the CABG group at one-year follow-up, the incidence of HF hospitalization was significantly higher in the former than that in the latter during the study period. During the acute phase, CABG may be associated with a higher risk of LV dysfunction and an increased incidence of sudden death. Indeed, a previous investigation focusing on patients with ischemic heart failure demonstrated that the monthly risk of sudden cardiac death after CABG was highest between the first and third months [25]. After the acute phase, CABG may be related to the benefits of a better alleviation of HF symptoms [26] as well as lower risks of revascularization compared to PCI [12,27,28]. The current study showed that the incidence of HF hospitalization was lower in patients receiving CABG than those undergoing PCI, especially in those with a SYNTAX score ≥33. The survival benefit of CABG for the patients with more complex coronary lesions stratified by the SYNTAX score was also reported by the other article [29].

### 4.3. The Improvement in LVEF and LV Reverse Remodeling after Revascularization

Significant improvement in post-procedural left ventricular function (i.e., LVEF > 50%) was noted in over half of the patients in patients receiving either CABG or PCI at three-year follow-up (61.5% vs. 52.9%, respectively) without notable difference between the two groups (*p* = 0.223). Nevertheless, a significantly higher prevalence of LVESV improvement >10% in the CABG group than that in the PCI group may highlight the clinical benefits of CABG in terms of enhancing LV reverse remodeling and reducing HF hospitalization when compared to patients undergoing PCI. The geometry plays a crucial role in different pathological mechanisms including restenosis caused by atherosclerosis and the risk of plaque rupture [30,31]. Additionally, the change in proximal arteries might influence the distal microcirculatory resistance, which is a quantitative evaluation of coronary microcirculatory function and provides a significant reference for the prediction, diagnosis, treatment, and prognosis of severe CAD [32,33]. Therefore, the underlying pathology needs to be further investigated in such population. LV remodeling improved the outcomes of HF hospitalization and the value of baseline LVEF appeared as independent predictor of improved LVEF after CABG [34].

### 4.4. Study Limitations

The present study had several limitations. First, this study was a retrospective study in which the CABG group mostly comprised patients with prior PCI failure. Second, to minimize bias from variations in procedural skills, the current study was a single-center investigation involving patients managed by a single intervention cardiologist and a single cardiovascular surgeon. Despite their seniority (both with clinical experience over 10 years), our findings may not be extrapolated to other cardiovascular teams. Third, since the study covered a time frame of over ten years, the evolution of skills and technology may bias the outcomes. Nevertheless, our findings still provided important information about LV improvement and the incidence of HF hospitalization after revascularization. Further large-scale studies are warranted to validate the results of the current study. 

## 5. Conclusions

The results of this study showed that, compared with PCI, revascularization with CABG was associated with a lower incidence of HF hospitalization but a worse survival outcome in patients with severe CAD and reduced EF. The correlation between CABG and a reduced incidence of HF hospitalization was more pronounced in patients with SYNTAX score ≥ 33.

## Figures and Tables

**Figure 1 diagnostics-12-02233-f001:**
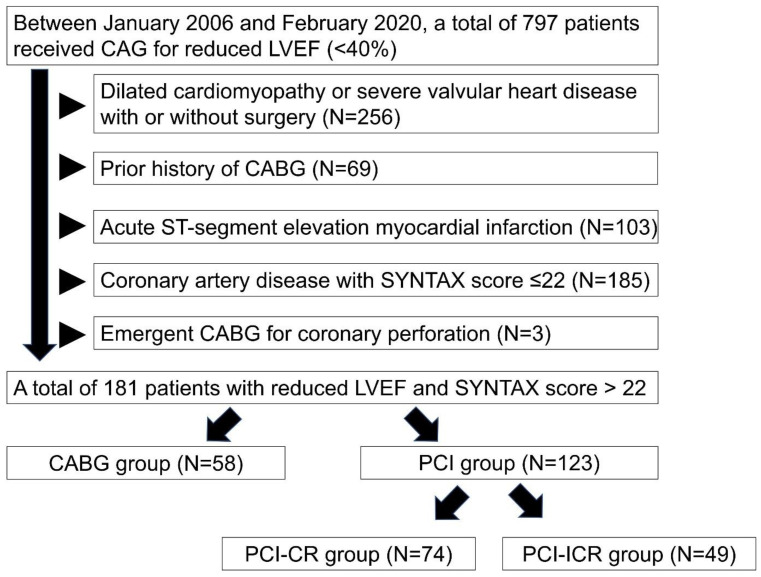
Flow chart of study population. Between 2006 February and 2020 February, a total of 797 patients received coronary angiograms due to left ventricular ejection fraction ≤40% in our hospital. We excluded patients of dilated cardiomyopathy or severe valvular heart disease with and without surgery (*n* = 256), prior history of coronary artery bypass grafting (CABG) (*n* = 69), acute ST-segment myocardial infarction (*n* = 103), coronary artery disease with low SYNTAX score ≤22 (*n* = 185) and emergent CABG for coronary perforation (*n* = 3). Percutaneous coronary intervention group was classified into complete revascularization (CR) group (PCI-CR) (*n* = 74) and incomplete-CR group (PCI-ICR) (*n* = 49). Abbreviation: CAG: coronary angiography; LVEF: left ventricular ejection fraction; CABG: coronary artery bypass grafting; PCI: percutaneous coronary intervention; CR: complete revascularization; ICR: incomplete revascularization.

**Figure 2 diagnostics-12-02233-f002:**
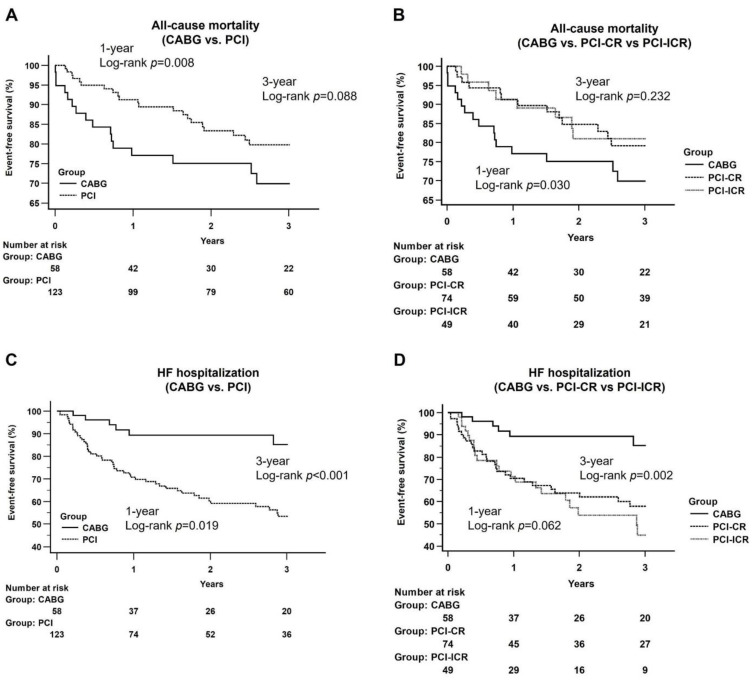
Kaplan–Meier curves of all-cause mortality and heart failure hospitalization between groups. (**A**) CABG vs. PCI: A higher incidence of all-cause mortality was noted at the one-year follow-up period (log-rank *p* = 0.008) and did not differ at the three-year follow-up period (log-rank *p* = 0.088); (**B**) CABG vs. PCI-CR vs. PCI-ICR: CABG group had a higher incidence of all-cause mortality at the one-year follow-up period (log-rank *p* = 0.030), and no significant difference was noted at the three-year follow-up period (log-rank *p* = 0.232). Abbreviation: CABG: coronary artery bypass grafting; PCI: percutaneous coronary intervention; CR: complete revascularization; ICR: incomplete revascularization; (**C**) CABG vs. PCI: CABG group had a lower incidence of HF hospitalization at one-year follow-up period (log-rank *p* = 0.019) and three-year follow-up period (log-rank *p* < 0.001); (**D**) CABG vs. PCI-CR vs. PCI-ICR: there was no significant difference at the one-year follow-up period (log-rank *p* = 0.062) and CABG group had a significantly lower incidence of HF hospitalization at three-year follow-up period (log-rank *p* = 0.002).

**Figure 3 diagnostics-12-02233-f003:**
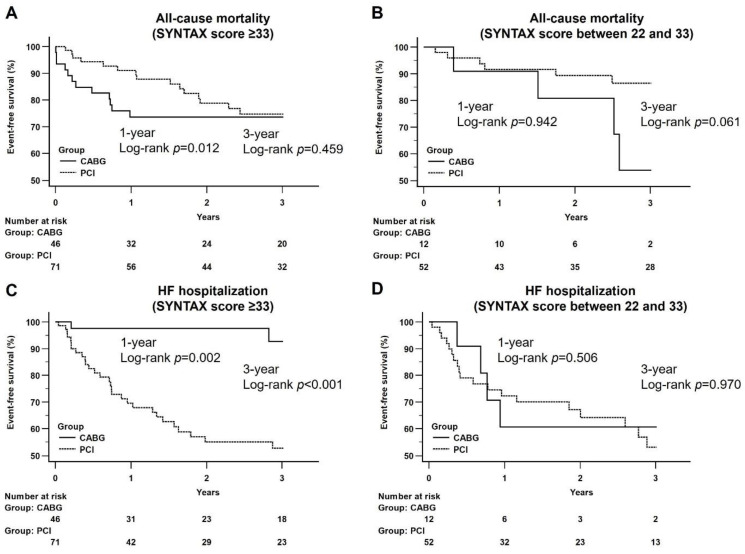
Kaplan–Meier curves of all-cause mortality and heart failure hospitalization at follow-up period between CABG and PCI groups by a different range of SYNTAX score. (**A**) CABG vs. PCI (SYNTAX score ≥ 33): A better survival outcome was noted at the one-year follow-up period (log-rank *p* = 0.012) and did not differ at the three-year follow-up period between CABG and PCI groups (log-rank *p* = 0.459); (**B**) CABG vs. PCI (SYNTAX score between 22 and 33): There was no significant difference at the one-year and three-year follow-up period between CABG and PCI groups.; (**C**) CABG vs. PCI (SYNTAX score ≥ 33): CABG group had a significantly lower incidence of HF hospitalization at the one-year (log-rank *p* = 0.002) and three-year (log-rank *p* < 0.001) follow-up period; (**D**) CABG vs. PCI (SYNTAX score between 22 and 33): There was no significant difference at the one-year and three-year follow-up period between CABG and PCI groups.

**Table 1 diagnostics-12-02233-t001:** Baseline characteristics and echocardiographic parameters of the study patients.

	CABG Group	PCI Group			*p* Value	*p* Value
		All Patients	CR	ICR	CABG vs. All PCI	CABG vs. CR
** *Number* **	58	123	74	49		
** *General demographics* **						
Age (years)	62 ± 9.0	66 ± 12.1	66 ± 12.1	66 ± 12.2	0.016 *	0.049 *
Male sex (%)	50 (86.2)	103 (83.7)	59 (79.7)	44 (89.8)	0.826	0.364
BMI (kg/m^2^)	25.1 ± 3.5	25.1 ± 4.4	24.5 ± 3.8	26.0 ± 5.2	0.938	0.909
** *Comorbidities* **						
Hypertension (%)	45 (77.6)	84 (68.3)	50 (67.6)	34 (69.4)	0.222	0.244
Diabetes mellitus (%)	36 (62.1)	77 (62.6)	45 (60.8)	32 (65.3)	1.000	1.000
PAOD (%)	2 (3.4)	10 (8.1)	3 (4.1)	7 (14.3)	0.343	1.000
COPD (%)	2 (3.4)	8 (6.5)	4 (5.3)	4 (8.2)	0.505	0.694
ESRD (%)	12 (20.7)	12 (9.8)	7 (9.5)	5 (10.2)	0.059	0.083
CKD stage ≥ 3 (%)	32 (55.2)	59 (48.0)	36 (48.6)	23 (46.9)	0.427	0.487
Smoking (%)	33 (56.9)	58 (47.2)	30 (40.5)	28 (57.1)	0.265	0.079
Prior PCI history (%)	27 (46.6)	24 (19.5)	13 (17.6)	11 (22.4)	<0.001 *	0.001 *
Prior MI < 90 days (%)	26 (44.8)	43 (35.0)	29 (39.2)	14 (28.6)	0.251	0.594
NYHA class ≥ 3 (%)	47 (81.0)	69 (56.1)	42 (56.8)	27 (55.1)	0.001 *	0.005 *
** *Clinical presentation* **					0.140	0.506
Acute coronary syndrome (%)	48 (82.8)	88 (71.5)	59 (79.7)	29 (59.2)		
Stable angina/HF (%)	10 (17.2)	35 (28.5)	15 (20.3)	20 (40.8)		
** *Laboratory examination* **						
Creatinine (exclude ESRD) (mg/dL)	1.20 (1.00–1.51)	1.17 (1.06–1.24)	1.15 (1.04–1.30)	1.19 (1.03–1.29)	0.402	0.666
** *New EuroSCORE II* **	6.25 (4.15–8.29)	2.73 (2.48–3.26)	2.81 (2.40–3.64)	2.73 (2.40–3.32)	<0.001 *	<0.001 *
** *Coronary complexity* **						
SYNTAX score	40.5 ± 9.8	35.4 ± 8.3	35.3 ± 7.9	35.7 ± 8.8	<0.001 *	0.001 *
SYNTAX score ≥ 33 (%)	46 (79.3)	71 (57.7)	45 (60.8)	26 (53.1)	0.005 *	0.025 *
Left main (%)	20 (34.5)	26 (21.1)	18 (24.3)	8 (16.3)	0.068	0.246
MVD					0.325	0.140
2-V-D (%)	9 (15.5)	28 (22.8)	20 (27.0)	8 (16.3)		
3-V-D (%)	49 (84.5)	95 (77.2)	54 (73.0)	41 (83.7)		
** *Mechanical support (%)* **	15 (25.9)	25 (20.3)	13 (17.6)	12 (24.5)	0.445	0.287
** *Medication* **						
ACEI/ARB/ARNI use (%)	37 (67.3)	102 (83.6)	61 (83.6)	41 (83.7)	0.018 *	0.037 *
β-blocker use (%)	35 (63.6)	107 (87.7)	66 (90.4)	41 (83.7)	<0.001 *	< 0.001 *
MRA (%)	28 (50.9)	33 (27.0)	21 (28.8)	12 (24.5)	0.003 *	0.017 *
Furosemide (%)	16 (27.6)	63 (51.2)	35 (47.3)	28 (57.1)	0.004 *	0.030 *
** *Echocardiographic parameters* **						
** *Baseline* **						
LA dimension (mm)	40.2 ± 6.3	41.4 ± 6.7	41.0 ± 6.0	42.1 ± 7.6	0.251	0.461
LVEF (%)	31.2 ± 6.8	31.6 ± 7.1	31.4 ± 7.1	31.8 ± 7.1	0.722	0.866
<30% (%)	26 (44.8)	39 (31.7)	25 (33.8)	14 (28.6)	0.098	0.212
LVEDV (ml)	183.4 ± 46.2	185.9 ± 55.7	187.8 ± 59.8	176.9 ± 57.4	0.772	0.541
LVESV (ml)	126.9 ± 40.6	126.8 ± 44.5	129.4 ± 40.6	121.7 ± 41.2	0.979	0.682
AR grade > 2 (%)	1 (1.7)	14 (11.4)	8 (10.8)	6 (12.2)	0.039 *	0.077
MR grade > 2 (%)	21 (36.2)	46 (37.4)	29 (39.2)	17 (34.7)	1.000	0.857
TR grade > 2 (%)	9 (15.5)	26 (21.1)	15 (20.3)	11 (22.4)	0.425	0.506
TRPG (mmHg)	25.0 (21.0–32.0)	26.5 (24.0–31.0)	26.5 (23.0–32.4)	26.5 (22.0–37.0)	0.767	0.961
** *Follow-up* **						
LA dimension (mm)	40.6 ± 7.3	39.7 ± 7.1	38.2 ± 7.3	42.0 ± 6.2	0.483	0.087
LVEF (%)	45.4 ± 13.8	42.8 ± 15.3	44.7 ± 15.2	39.8 ± 15.3	0.288	0.705
>50% (%)	22 (42.3)	37 (36.3)	23 (37.7)	12 (29.3)	0.169	0.701
>40% (%)	32 (61.5)	54 (52.9)	34 (55.7)	18 (43.9)	0.233	0.570
LVEDV (ml)	169.0 ± 67.3	178.7 ± 64.6	177.4 ± 67.3	180.7 ± 61.3	0.386	0.510
LVESV (ml)	97.7 ± 58.9	108.0 ± 57.9	104.8 ± 60.3	112.7 ± 54.5	0.300	0.527
AR grade > 2 (%)	0 (0)	8 (6.5)	3 (4.9)	5 (12.2)	0.052	0.248
MR grade > 2 (%)	7 (13.5)	15 (12.2)	10 (16.4)	5 (12.2)	1.000	0.794
TR grade > 2 (%)	5 (9.6)	10 (8.1)	5 (8.2)	5 (12.2)	1.000	1.000
TRPG (mmHg)	22.0 (19.0–24.0)	23.0 (20.0–25.0)	22.5 (19.0–24.7)	25.0 (17.7–33.3)	0.583	0.303
** *The change of left ventricular volume between baseline and follow-up* **						
Reducing LVEDV > 10% (%)	25 (48.1)	45 (44.1)	29 (47.5)	16 (39.0)	0.733	1.000
Reducing LVESV > 10% (%)	35 (67.3)	51 (50.0)	32 (52.5)	19 (46.3)	0.041 *	0.127
** *The change of LVEF between baseline and follow-up* **	13.5 (7.9–20.1)	10.5 (5.0–17.4)	13.0 (6.0–20.6)	6.0 (−1.2–14.2)	0.233	0.675
Improving mean LVEF > 10% (%)	32 (61.5)	54 (52.9)	35 (57.4)	19 (46.3)	0.391	0.703
** *F/U duration (days)* **	846 (563–1257)	1052 (834–1187)	1145 (838–1269)	874 (674–1161)	0.762	0.799

Data are expressed as mean ± standard deviation or as median and interquartile range or as number (percentage). Abbreviation: PCI: percutaneous coronary intervention; CABG: coronary artery bypass grafting; CR: complete revascularization; ICR: incomplete revascularization; PAOD: peripheral arterial occlusive disease; COPD: chronic obstructive pulmonary disease; CKD: chronic kidney disease; MI: myocardial infarction; NYHA: New York Heart Association; ESRD: end-stage renal disease; SYNTAX: Synergy between PCI with Taxus and Cardiac Surgery; MVD: microvascular disease; MRA: Magnetic Resonance Angiography; LA: left atrium; LV: left ventricle; LVEF: left ventricular ejection fraction; LVEDV: LV end-diastolic volume; LVESV: LV end-systolic volume; AR: aortic regurgitation; MR: mitral regurgitation; TR: tricuspid regurgitation; TRPG: tricuspid regurgitation peak gradient; F/U: follow up. *: statistically significant.

**Table 2 diagnostics-12-02233-t002:** Clinical outcomes of the study patients.

	CABG Group	PCI Group			*p* Value	*p* Value
		All Patients	CR	ICR	CABG vs. All PCI	CABG vs. CR
** *Number* **	58	123	74	49		
** *In-hospital course* **						
In-hospital mortality (%)	3 (5.2)	1 (0.8)	1 (1.4)	0 (0)	0.098	0.319
AKI (%)	4 (8.7)	17 (15.3)	9 (13.4)	8 (18.2)	0.315	0.555
Post-procedural HD (%)	3 (6.5)	5 (4.5)	4 (6.0)	1 (2.3)	0.693	1.000
** *One-year follow-up duration* **						
Recurrent MI (%)	2 (4.3)	10 (8.8)	6 (8.8)	4 (8.9)	0.512	0.469
Revascularization (%)	2 (3.4)	12 (9.8)	6 (8.1)	6 (12.2)	0.231	0.465
Sudden death/Ventricular arrhythmia (%)	7 (13.5)	3 (2.6)	1 (1.5)	2 (4.3)	0.011 *	0.021 *
HF hospitalization (%)	5 (11.1)	33 (28.2)	20 (28.2)	13 (28.3)	0.023 *	0.037 *
Stroke (%)	0 (0)	0 (0)	0 (0)	0 (0)	-	-
CV mortality (%)	11 (19.6)	6 (5.0)	4 (5.6)	2 (4.3)	0.005 *	0.024 *
All-cause mortality (%)	13 (22.4)	10 (8.1)	6 (8.1)	4 (8.2)	0.015 *	0.025 *
** *Three-year follow-up duration* **						
Recurrent MI (%)	4 (9.1)	12 (11.7)	7 (11.3)	5 (12.2)	0.778	1.000
Revascularization (%)	4 (6.9)	27 (22.0)	15 (20.3)	12 (24.5)	0.011 *	0.044 *
Sudden death/Ventricular arrhythmia (%)	10 (19.2)	9 (8.3)	5 (7.7)	4 (9.1)	0.044 *	0.094
HF hospitalization (%)	6 (13.6)	48 (42.5)	27 (39.7)	21 (46.7)	0.001 *	0.003 *
Stroke (%)	1 (1.7)	1 (0.8)	1 (1.4)	0 (0)	0.539	1.000
CV mortality (%)	14 (25.0)	13 (11.4)	10 (14.1)	3 (7.0)	0.027 *	0.170
All-cause mortality (%)	16 (27.6)	21 (17.1)	13 (17.6)	8 (16.3)	0.116	0.205

Data are expressed as mean ± standard deviation or as number (percentage). Abbreviation: PCI: percutaneous coronary intervention; CABG: coronary artery bypass grafting; HF: heart failure; CV: cardiovascular; CR: complete revascularization; ICR: incomplete revascularization. *: statistically significant.

**Table 3 diagnostics-12-02233-t003:** Hazard ratio of 3-year HF hospitalization when PCI compared to CABG.

*Variable*	Hazard Ratio	*p* Value	95% CI
PCI (whole group)	3.868	0.002 *	1.655–9.042
PCI with CR	3.553	0.005 *	1.466–8.610
CABG	1.000		
** *Syntax score ≥ 33* **			
PCI	10.120	0.002 *	2.412–42.451
CABG	1.000		
** *Syntax score between 22 and 33* **			
PCI	0.925	0.888	0.314–2.727
CABG	1.000		

Abbreviation: HF: heart failure; PCI: percutaneous coronary intervention; CABG: coronary artery bypass grafting; CI: confidence interval; CR: complete revascularization; SYNTAX: Synergy between PCI with Taxus and Cardiac Surgery. *: statistically significant.

## Data Availability

The data was available when the request to the corresponding author.

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
