# Peer review of "Coronary Artery Bypass Graft Surgery Brings Better Benefits to Heart Failure Hospitalization for Patients with Severe Coronary Artery Disease and Reduced Ejection Fraction"

_diagnostics, 2022, doi:10.3390/diagnostics12092233_

Round 1
Reviewer 1 Report
Please kindly find attached review comments.

Author Response
Specific responses to the first reviewer’s comments:
Reviewer #1:
General and specific comments to the authors
In this paper, the authors compared the outcomes between percutaneous coronary intervention (PCI) and coronary artery bypass graft surgery (CABG) for revascularization in patients with reduced ejection fraction (EF) and severe coronary artery disease (CAD). Overall, it is very well written and structured. However, there are some flaws in statistical analysis, and a lack of in-depth discussion, which need to be improved to meet publishable standards
Comment 1: The statistical analysis need further improvemen Normality test is needed before t-test. If the data do not follow normal distribution, the non-parametric test need to be considered.
Responses: We have revised our manuscript according to this comment in the “Methods” paragraph, in page 3, paragraph 3, lines 109-110 of the revised manuscript. We also corrected the results and tables.
“Continuous variables are presented as mean ± standard deviation (SD) and categorical data are expressed as absolute values with percentages for normally distributed parameters and showed median and interquartile range for non-normally distributed parameters.”
Comment 2: The Chi -squared test also has its premi If the conditions are violated, the Fisher's exact test need to be performed. Please renew the statistical methods and results.
Responses: We have revised our manuscript according to this comment in the “Methods” paragraph, in page 3, paragraph 3, line 112 of the revised manuscript.
“Baseline patient characteristics …… a Chi-square test or Fisher's exact test for categorical variables.”
Comment 3: Please mark the statistically significant (*) values in the tabl
Responses: We have added the mark (*) for statistical significance in tables.
Comment 4: The results deserve an in-depth discussion from a pathophysiological vi The geometry plays a key role in different pathological mechanisms including restenosis caused by the development of atherosclerosis (Refer: 10.1088/2057-1976/aa9a09), as well as the risk of plaque rupture (Refer: 10.1016/j.jacc.2007.10.030). Additionally, the distal microcirculatory resistance might be influenced by the change of proximal arteries. Therefore, the underlying pathology need to be further investigated.
Responses: We have revised our manuscript according to this comment in the “Discussion” paragraph, in page 12, paragraph 1, lines 345-353 of the revised manuscript. We also added the references.
“The geometry plays a crucial role in different pathological mechanisms including restenosis caused by atherosclerosis and the risk of plaque rupture [30, 31]. Additionally, the change in proximal arteries might influence the distal microcirculatory resistance, which is a quantitative evaluation of coronary microcirculatory function and provides a significant reference for the prediction, diagnosis, treatment, and prognosis of severe CAD [32, 33]. Therefore, the underlying pathology needs to be further investigated in such population. LV remodeling improved the outcomes of HF hospitalization and the value of baseline LVEF appeared as independent predictor of improved LVEF after CABG [34].”
Thank you for your constructive and valuable comments.
Reviewer 2 Report
I have read the manuscript with great interest.
I think the intro section is quite superficial. I would see deeper intro including the following:
Translational Research in Anatomy
Volume 22, January 2021, 100083
Cardiac ultrasound: An Anatomical and Clinical Review
Islam Aly, Asad Rizvi, Wallisa Roberts, Shehzad Khalid, Mohammad W.Kassem, Sonja Salandy, Maira du Plessis, R. ShaneTubbs, Marios Loukas
https://doi.org/10.1016/j.tria.2020.100083
Factors causing variability in formation of coronary collaterals during coronary artery disease
S. Balakrishnan, B. Senthil Kumar
Folia Morphologica 2022 Ahead of Print
Morphologic comparison of blood vessels used for coronary artery bypass graft surgery
M. Garnizone, E. Vartina, M. Pilmane
Folia Morphologica Vol 81, No 3 (2022)
Prevalence of congenital coronary artery anomalies and variants in 726 consecutive patients based on 64-slice coronary computed tomography angiography
K. Szymczyk, M. Polguj, E. Szymczyk, A. Majos, P. Grzelak, L. Stefańczyk
Folia Morphologica Vol 73, No 1 (2014)
The readers need more clear inclusion and exclusion criteria. Were the statistics conducted following the PRISMA accordances?
I would suggest to add :
Consensus guidelines for the uniform reporting of study ethics in anatomical research within the framework of the anatomical quality assurance (AQUA) checklist
Henry, BMHenry, Brandon Michael) [1] , [2] ; Vikse, Jens) [1] , [2] ; Pekala, Przemyslaw) [1] , [2] ; Loukas, Marios) [3] ; Tubbs, R. Shane) [4] ; Walocha, Jerzy A.) [1] , [2] ; Jones, D. Gareth) [5] ; Tomaszewski, Krzysztof A.) [1] , [2]
Clinical Anatomy, Volume 31
Issue
4
Page
521-524
DOI
10.1002/ca.23069
Published
MAY 2018
The discussion refers to the most important sources but I suppose there are lots of newer articles. Can be improved.
Author Response
Specific responses to the second reviewer’s comments:
Reviewer #2:
General and specific comments to the authors
I have read the manuscript with great interest.
Comment 1: I think the intro section is quite superficial. I would see deeper intro including the following:
Translational Research in Anatomy Volume 22, January 2021, 100083 Cardiac ultrasound: An Anatomical and Clinical Review Islam Aly, Asad Rizvi, Wallisa Roberts, Shehzad Khalid, Mohammad W.Kassem, Sonja Salandy, Maira du Plessis, R. ShaneTubbs, Marios Loukas https://doi.org/10.1016/j.tria.2020.100083
Factors causing variability in formation of coronary collaterals during coronary artery disease
- Balakrishnan, B. Senthil Kumar
Folia Morphologica 2022 Ahead of Print
Morphologic comparison of blood vessels used for coronary artery bypass graft surgery
- Garnizone, E. Vartina, M. Pilmane
Folia Morphologica Vol 81, No 3 (2022)
Prevalence of congenital coronary artery anomalies and variants in 726 consecutive patients based on 64-slice coronary computed tomography angiography
- Szymczyk, M. Polguj, E. Szymczyk, A. Majos, P. Grzelak, L. Stefańczyk
Folia Morphologica Vol 73, No 1 (2014)
Responses: We have revised our manuscript according to this comment in the “Background” paragraph, in page 2, paragraph 1, lines 49-53 of the revised manuscript. We also added the references.
“In such population, the evaluation of ischemic area and involved vessels and the healthy of collateral vessel are very important and the methods of PCI or CABG also have the impact on the outcomes [13-16]. The decompensated status of HF with reduced EF and post-PCI or CABG stunning cardiomyopathy also plays an important role of this issue.”
Comment 2: The readers need more clear inclusion and exclusion criteria. Were the statistics conducted following the PRISMA accordances?
I would suggest to add :
Consensus guidelines for the uniform reporting of study ethics in anatomical research within the framework of the anatomical quality assurance (AQUA) checklist
Henry, BM (Henry, Brandon Michael) [1] , [2] ; Vikse, J (Vikse, Jens) [1] , [2] ; Pekala, P (Pekala, Przemyslaw) [1] , [2] ; Loukas, M (Loukas, Marios) [3] ; Tubbs, RS (Tubbs, R. Shane) [4] ; Walocha, JA (Walocha, Jerzy A.) [1] , [2] ; Jones, DG (Jones, D. Gareth) [5] ; Tomaszewski, KA (Tomaszewski, Krzysztof A.) [1] , [2]
Clinical Anatomy, Volume 31 Issue 4 Page 521-524 DOI 10.1002/ca.23069 Published MAY 2018
Responses: We have revised our manuscript according to this comment in the “Methods” paragraph, in page 2, paragraph 4, lines 64-81 of the revised manuscript.
“Between February 2006 and February 2020, a total 797 patients with HF with reduced EF received coronary angiography were recruited. The medical records of all patients (age>18) with severe heart failure (defined as left ventricular EF ≤40%) regardless of gender undergoing either PCI by a single intervention cardiologist or CABG by a single cardiovascular surgeon with pre-procedural coronary angiograms …… were recorded. Patients with (1) dilated cardiomyopathy or severe valvular heart disease with and without surgery, (2) prior history of CABG, (3) acute ST-segment myocardial infarction, (4) CAD with a SYNTAX score ≤22, and (5) emergent CABG for coronary perforation were excluded from the present study. The enrolled patients were severe CAD (SYNTAX score >22), who need PCI or CABG for ischemic cardiomyopathy and HF with reduced EF. The clinical condition, and vascular characteristics, and echocardiographic data were compared between groups. PCI were classified into two subgroups: complete revascularization (PCI-CR) and incomplete revascularization (PCI-ICR) to evaluate the effect of CR.”
Comment 3: The discussion refers to the most important sources but I suppose there are lots of newer articles. Can be improved.
Responses: We have revised our manuscript according to this comment in the “Discussion” paragraph, in page 1, paragraph 1, lines 295-307 of the revised manuscript. We also added the references.
“A previous cohort study including 3584 patients with three-vessel and/or left main disease demonstrated a higher incidence of readmission for heart failure in patients with impaired LV systolic function (i.e., LVEF < 50%) undergoing PCI than that in those receiving CABG (24% vs. 15%, Hazard Ratio = 2.22, p<0.01). However, there was no significant difference in the incidence of readmission between the PCI and CABG groups (8% vs. 6%, respectively; Hazard Ratio = 1.39, p=0.11) for patients with preserved LV systolic function (i.e., LVEF < 50%) receiving either of the two procedures [17]. Another recent study including 4794 patients with reduced LVEF also reported consistent results (PCI vs. CABG = 25.8% vs. 20.1%, Hazard Ratio = 1.5, p<0.01) [9]. Therefore, the findings of those studies were consistent with those of the present study (PCI vs. CABG; 42.5% vs. 13.6%; Hazard Ratio = 3.868, p=0.002). In one metaanalysis of severely reduced EF, CABG decreases the risk of mortality, MI, and revascularization than PCI, but may increase the short-term and long-term risk of stroke [20].”
Thank you for your constructive and valuable comments.
Round 2
Reviewer 1 Report
Thanks for the update. The authors have much improved the manuscript. Most of the issues in my earlier comments have been addressed. However, there are still some minor issues, e.g., different formats of the references. I suggest the authors to double check these details and ask a professional native speaker for proofreading.
Reviewer 2 Report
All my comments were inckuded. Thank you